# Laboratory generation of new parthenogenetic lineages supports contagious parthenogenesis in *Artemia*

Marta Maccari[1,2], Francisco Amat[1], Francisco Hontoria[1] and Africa Gómez[2]

[1] Instituto de Acuicultura de Torre de la Sal (Consejo Superior de Investigaciones Científicas), Ribera de Cabanes (Castellón), Spain
[2] School of Biological, Biomedical and Environmental Sciences, University of Hull, Hull, United Kingdom

## ABSTRACT

Contagious parthenogenesis—a process involving rare functional males produced by a parthenogenetic lineage which mate with coexisting sexual females resulting in fertile parthenogenetic offspring—is one of the most striking mechanisms responsible for the generation of new parthenogenetic lineages. Populations of the parthenogenetic diploid brine shrimp *Artemia* produce fully functional males in low proportions. The evolutionary role of these so-called *Artemia* rare males is, however, unknown. Here we investigate whether new parthenogenetic clones could be obtained in the laboratory by mating these rare males with sexual females. We assessed the survival and sex ratio of the hybrid ovoviviparous offspring from previous crosses between rare males and females from all Asiatic sexual species, carried out cross-mating experiments between F1 hybrid individuals to assess their fertility, and estimated the viability and the reproductive mode of the resulting F2 offspring. Molecular analysis confirmed the parentage of hybrid parthenogenetic F2. Our study documents the first laboratory synthesis of new parthenogenetic lineages in *Artemia* and supports a model for the contagious spread of parthenogenesis. Our results suggest recessive inheritance but further experiments are required to confirm the likelihood of the contagious parthenogenesis model.

## INTRODUCTION

Parthenogenesis in animals has evolved through different molecular mechanisms that influence the initial genetic variability of parthenogenetic strains and therefore have important implications on their evolutionary success and persistence (*Simon et al., 2003*). One of the most striking mechanisms responsible for the generation of new parthenogenetic lineages is contagious parthenogenesis (*Simon et al., 2003*; *Schön, Martens & van Dijk, 2009*). This involves a parthenogenetic lineage able to produce functional males, which mate with coexisting sexual females producing fertile parthenogenetic hybrid offspring. These new parthenogenetic lineages will combine genetic diversity of

Corresponding author
Marta Maccari,
martamaccari@gmail.com

the maternal sexual species and their paternal parthenogenetic ancestor, including the genetic fragments linked to the parthenogenesis (*Simon et al., 2003*; *Tucker et al., 2013*). This mechanism has been documented in aphids and parasitoid wasps (*Schneider et al., 2002*; *Sandrock & Vorburger, 2011*; *Delmotte et al., 2013*), and most extensively in the *Daphnia pulex* species complex (*Innes & Hebert, 1988*; *Paland, Colbourne & Lynch, 2005*). In North American *D. pulex* parthenogenetic lineages, at least two distinct unrecombined haplotypes on chromosome VIII and IX are implied in the sex-limited meiosis suppression (*Lynch et al., 2008*; *Eads et al., 2012*; *Tucker et al., 2013*). These haplotypes leading to obligate parthenogenesis in *D. pulex* stem from a single recent event of hybridization with its sister taxon *D. pulicaria* (*Xu et al., 2013*; *Tucker et al., 2013*). Multiple new parthenogenetic lineages have arisen since this event as males produced by asexual lineages spread these parthenogenesis-inducing haplotypes by mating with sexual females.

*Artemia*, an anostracan branchiopod commonly known as brine shrimp, is a typical inhabitant of hypersaline inland lakes and coastal lagoons and salterns. This genus includes sexual species and lineages of obligate parthenogenetic populations of diverse ploidy levels (*Abatzopoulos, 2002*), which makes it a good model system to investigate evolutionary transitions between reproductive systems. Parthenogenetic populations are restricted to the Old World where they co-occur with several sexual species in sympatry in various areas (*Abatzopoulos, 2002*; *Agh et al., 2007*; *Abatzopoulos et al., 2009*; *Maccari et al., 2013*). All strains of *Artemia* can reproduce either ovoviviparously, with the release of free-swimming nauplii broods when they complete their development in the ovisac (therefore, without a dormant phase), or oviparously with the production of broods of diapausing cysts (*Browne, 1980*; *Abatzopoulos, 2002*).

In *Artemia*, both sexual and asexual females are heterogametic (ZW) (*Stefani, 1963*; *Bowen, 1963*; *Bowen, 1965*; *De Vos et al., 2013*). Diploid parthenogenetic lineages reproduce through automictic parthenogenesis, although the cytological details are controversial (*Cuellar, 1987*). It appears that diploidy restoration results in female offspring genetically identical to the mother barring mutation or recombination (*Abreu-Grobois, 1987*; *Stefani, 1960*). Parthenogenetic diploid *Artemia* populations produce fully functional males in low proportions (*Stefani, 1964*; *Bowen et al., 1978*; *MacDonald & Browne, 1987*; *Maccari et al., 2013*). *Abreu-Grobois & Beardmore (2001)* showed that rare males remain heterozygous at the same allozyme loci as their mothers, suggesting that rare males are produced as a result of rare ZW recombination events. These 'rare males' can generate viable offspring when crossed with females of sexual Asiatic species (*Bowen et al., 1978*; *Cai, 1993*; *Maccari et al., 2013*), to which they are closely related genetically (*Muñoz et al., 2010*; *Maniatsi et al., 2011*; *Maccari, Amat & Gómez, 2013*), but they are reproductively isolated with other more distantly related species (*MacDonald & Browne, 1987*). However, the evolutionary role of rare males in the generation of *Artemia* parthenogenetic lineages is unknown (*Maccari et al., 2013*). The occurrence of contagious parthenogenesis has been suggested in light of the polyphyletic nature of maternal diploid parthenogenetic lineages (*Maccari, Amat & Gómez, 2013*), but we do not know if rare males are able to transmit parthenogenesis to their offspring, a requisite for contagious parthenogenesis. In an early study,

*Bowen et al. (1978)* crossed two parthenogenetic rare males, one from Yamaguchi (Japan) and the other one from Madras (India), with one sexual female of *A. urmiana* and one *A. franciscana* respectively, and concluded that parthenogenetic reproduction could not be transmitted through males because they failed to obtain parthenogenetic offspring either in hybrid F1, F2 or F2 backcross.

Laboratory generation and establishment of unisexual lineages can be a useful tool to complement phylogenetic approaches to identify the mechanism involved in the transition from sexual to parthenogenetic reproduction. However, most laboratory hybrids often exhibit low fertility and survival, or show deformation and abnormalities (*Vrijenhoek, 1989*; *Mantovani et al., 1996*). In vertebrates, the first successful laboratory generation of a unisexual hybrid involved the origin of the hybridogenetic fish *Poeciliopsis monacha-lucida* through crosses of *P. monacha* females with *P. lucida* males (*Schultz, 1973*). Laboratory hybrids of hemiclonal European water frog *R. esculenta* (*Rana ridibunda* x *Rana lessonae*) show faster larval growth, earlier metamorphosis, and higher resistance to hypoxic conditions than their parental species and the equivalent hybrids in nature (*Hotz et al., 1999*). More recently, *Lutes et al. (2011)* generated self-sustaining tetraploid lineages of parthenogenetic lizards by pairing males of diploid sexual species *Aspidoscelis inornata* with females of the triploid parthenogenetic species *Aspidocelis exsanguis*. In invertebrates, the first laboratory generation of clonal hybrids in *D. pulex* was obtained by crossing males from obligately parthenogenetic clones with cyclically parthenogenetic females (*Innes & Hebert, 1988*). In addition, new lineages of thelytokous parthenogenetic lineages have been obtained in the wasp *Lysiphlebus fabarum* and in a South African honeybee, *Apis mellifera capensis* (*Lattorff, Moritz & Fuchs, 2005*; *Sandrock & Vorburger, 2011*).

Here we assess the reproductive role of rare males and investigate whether new parthenogenetic clones could be produced in the laboratory as support for the contagious origin of parthenogenetic lineages in *Artemia*. For this purpose, (1) we assess the survival and sex ratio of the hybrid ovoviviparous offspring obtained from the previous crosses from *Maccari et al. (2013)* between rare males and four Asiatic sexual species, (2) we carry out cross-mating experiments between these F1 hybrid individuals to assess their fertility, (3) we estimate the viability and the reproductive mode of the resulting F2 offspring; (4) finally we demonstrate genetically that parthenogenetic F2 are indeed the descendants of the original crosses. This study shows that *Artemia* has the potential of generating parthenogenetic strains through contagious parthenogenesis.

## MATERIALS AND METHODS

### Populations and mating experiments

In a previous study, we set up mating experiments between rare males from the diploid parthenogenetic *Artemia* population from Bagdad (Iraq, hereafter PD) and sexual females from Asiatic *Artemia* species to assess the fertility and the reproductive potential of rare males (*Maccari et al., 2013*). The females used were from the sexual Asiatic populations, *A. urmiana* from Koyashskoe Lake (Ukraine, URM), *A. sinica* from Yuncheng Lake (China, SIN), *A. tibetiana* from Lagkor Co Lake (Tibet, TIB) and *Artemia*

sp. from Kazakhstan (*Artemia* Reference Center code – ARC1039, unknown locality, KAZ). These interspecific crosses resulted in viable ovoviviparous and oviparous F1 offspring with similar or higher viability than controls (intraspecific sexual crosses) (*Maccari et al., 2013*).

## Survival rate, sex ratio and reproductive performance of hybrid generations

For this study, live nauplii obtained from each ovoviviparous F1 hybrid brood were reared separately in jars containing brine at 80 gL$^{-1}$ salinity, kept at 20–24 °C under mild aeration at a 12D:12L photoperiod and fed a mixture of *Dunaliella* sp and *Tetraselmis* sp. (1:1) microalgae every other day. When animals showed signs of reproductive maturity they were counted and sexed to estimate survival rates (the proportion of F2 offspring per pair attaining adulthood) and sex ratio (the proportion of males in the F2 offspring per pair). For this procedure the animals were placed in Petri dishes with seawater and anaesthetized with a few drops of freshwater saturated with chloroform and examined carefully under a binocular microscope. We tested for deviations from a 50% sex ratio per cross and per pair using a Chi-square goodness of fit test (Pearson's statistic) (*Wilson & Hardy, 2002*). Statistical analyses were performed with SPSS v. 15.0 (SPSS Inc., Chicago, USA).

Reproductive performance of the F1 hybrid individuals was evaluated in F1 × F1 cross fertility tests. For this purpose, 24 randomly size-matched hybrid F1 male–female pairs from each cross were transferred into separate small glass beakers (60 ml) under the culture conditions described above. Lifetime quantitative and qualitative reproductive outputs of each pair were monitored every other day during culture medium renewal events. For each paired F1 female we counted the number of unfertilized and fertilized broods, distinguishing the latter in oviparous and ovoviviparous broods. Eggs from unfertilised broods were identified as they are all smaller and white. In ovoviviparous offspring we also recorded the number of live and dead nauplii, and the number of abortive embryos (pale yellow-orange eggs). When oviparous offspring was produced, we counted the number of normally shelled diapausing cysts (pale grainy surface floating in 200 gl$^{-1}$ brine), as opposed to abortive, abnormally shelled embryos (bright brown colour cysts sinking in 200 gl$^{-1}$ brine) (*Maccari et al., 2013*).

Emerged F2 hybrid nauplii were reared until maturity as described above. They were counted and sexed to estimate their survival rate and sex ratio in the F2 generation. Then, males and females were individually isolated in containers until their deaths to check if females could reproduce in isolation, as would be expected in parthenogenetic individuals. It is possible that some parthenogenetic females could be sterile; in this case, our procedure will underestimate the frequency of parthenogenesis. The proportion of parthenogenetic female offspring produced in each cross was tested against the expectations of 25% if governed by a recessive allele in a single gene using a Chi-square goodness of fit test. In addition, to test whether the different crosses produced the same percentage of parthenogenetic female offspring we used a Chi-square homogeneity test.

 

## Paternity analysis of parthenogenetic F2 individuals

(a) *Microsatellite analysis*

The F2 hybrid generation resulting from crosses between rare males and sexual females from *A. urmiana* and *Artemia* sp. from Kazakhstan included parthenogenetic individuals. In order to rule out contamination and confirm that they were F2 individuals resulting from the original crosses, we screened three microsatellite loci, previously screened in the parental individuals in another study (*Maccari et al., 2013*), in the parthenogenetic F2 animals obtained. Each microsatellite locus (Apdq02TAIL, Apdq03TAIL and Apd05TAIL) (*Muñoz et al., 2008*) was amplified separately in PCRs performed as described in *Maccari et al. (2013)*. Alleles were scored using the CEQ Fragment Analysis software (Beckman Coulter[TM]) and checked manually. If F2 individuals had a paternal allele in any of the loci this would confirm that they were descendants of the diploid parthenogenetic rare males.

(b) *Maternal lineage*

The F2 resulting from the rare male x sexual female cross and F1 × F1 cross should carry the maternal DNA of the sexual strain. To establish the maternal lineage of the parthenogenetic F2 offspring, a 709-bp fragment of mitochondrial cytochrome *c* oxidase subunit I (COI) gene region was amplified in the parental (F0) individuals, in the F1 offspring and in the parthenogenetic F2 individuals. Total DNA was extracted and PCR was carried out as described previously (*Maccari et al., 2013*). PCR amplifications were sent to MACROGEN for sequencing, and the resulting electrophoregrams were checked by eye using CodonCode Aligner v. 3.5 (CodonCode Corporation, Dedham, MA).

# RESULTS

## Survival rate and sex ratio of F1 hybrid offspring

A total of 102 ovoviviparous hybrid F1 broods produced by the crosses between each combination of sexual species with rare males (*Maccari et al., 2013*) were reared to maturity. The live nauplii obtained in each brood were morphologically normal. Survival rates to adulthood were over 50% in all F1 hybrid offspring (Fig. 1), and were highest in the F1 PD × SIN (80%), and lowest in F1 PD × URM and F1 PD × TIB (ca. 56%)(for the codes of the hybrid crosses see Fig. 1). The overall mean sex ratio of F1 offspring across pairs ranged from 49% males in F1 PD × KAZ cross to 53% males in F1 PD × TIB cross and did not significantly differ from 50% in any cross (Fig. 1).

## Reproductive performance of F1 hybrid offspring

Prior to setting up the crosses, all females were isolated from males for two weeks to ensure that they could not reproduce in isolation (i.e., they were sexual females). No F1 females were able to reproduce when isolated from males. Then, a total of 24 mating pairs (F1 hybrid female × F1 hybrid male) were set up for each F1 produced in each combination of sexual species with rare males. As some individuals died before mating, the final number of experimental pairs ranged from 10 to 22 per cross, which produced a total of 173 fertile and 92 infertile F2 hybrid broods (Table 1). Ovoviviparous and oviparous F2 offspring viability is shown in Fig. 2. The percentage of abortive embryos was high in all crosses (between

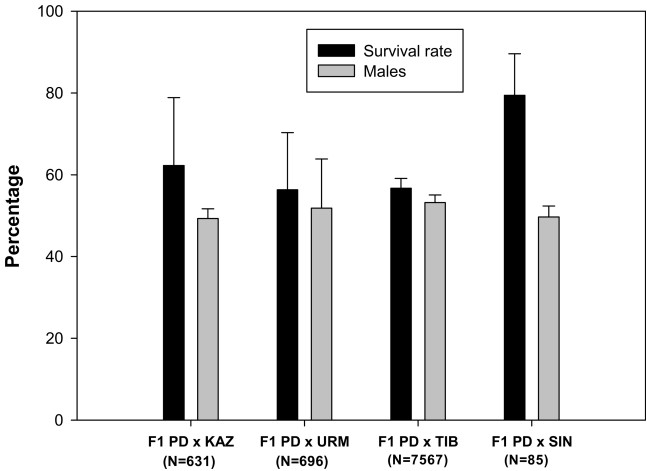

**Figure 1 Survival rate and sex ratio (overall percentage of males) in the F1 hybrid offspring from *Artemia* rare males and Asiatic sexual females.** F1 hybrids are from parental crosses between *Artemia urmiana* (URM), *A. sinica* (SIN), *A. tibetiana* (TIB), *Artemia* sp. from Kazakhstan (KAZ) and diploid parthenogenetic *Artemia* rare males (PD). Error bars are standard deviations.

**Table 1 Number of total, fertilized, ovoviviparous and oviparous broods in F1 *Artemia* hybrid offspring.** F1 hybrids are from parental crosses between *Artemia urmiana* (URM), *Artemia sinica* (SIN), *Artemia tibetiana* (TIB), *Artemia* sp. from Kazakhstan (KAZ) and diploid parthenogenetic *Artemia* rare males (PD).

| Cross | Pairs | Total broods | Fertilized broods | Ovoviviparous broods | Oviparous broods |
|---|---|---|---|---|---|
| F1 PD × KAZ | 18 | 80 | 42 | 37 | 5 |
| F1 PD × URM | 16 | 48 | 26 | 22 | 4 |
| F1 PD × TIB | 10 | 33 | 18 | 4 | 14 |
| F1 PD × SIN | 22 | 104 | 87 | 40 | 47 |

70% and 90%), while the proportion of live nauplii in all hybrid ovoviviparous broods was low (from 5% to 25%). In oviparous broods, the proportion of properly shelled cysts ranged from 25% in F2 PD × TIB to 61% in F2 PD × URM.

## Survival rate and sex ratio of F2 hybrid offspring

A total of 103 F2 ovoviviparous broods were recorded (Table 1), of which 35 broods from 27 pairs, characterized by the greatest number of nauplii, were followed to assess the survival rate and the sex ratio of the F2 offspring. F2 nauplii were morphologically normal but they had low survival rates when compared to F1 nauplii (Fig. 3). No F2 hybrid offspring produced by the crosses between rare male and *A. tibetiana* survived to maturity. The F2 PD × KAZ had the highest survival rate, about 37%, followed by the F2 PD × SIN (34%) and F2 PD × URM (24%). The overall mean sex ratio across pairs was significantly female-biased in F2 PD × KAZ and F2 PD × URM crosses (12% and 22% of

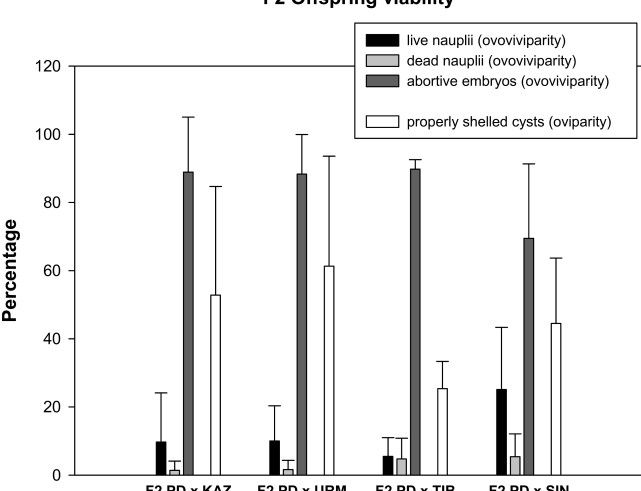

**Figure 2 Reproductive traits (offspring quantity and quality) in F2 hybrids between *Artemia* rare males and Asiatic sexual females.** The viability of ovoviviparous and oviparous broods is shown. Error bars are standard deviations.

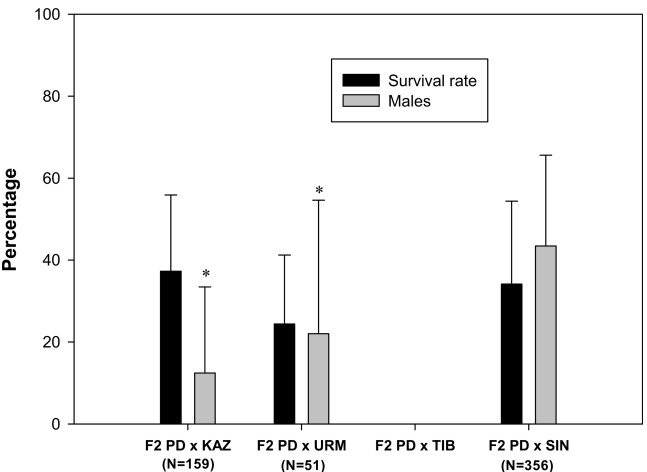

**Figure 3 Survival rate and sex ratio (overall percentage of males) in the F2 hybrid offspring from *Artemia* rare males and Asiatic sexual females.** F2 hybrids are from crosses between F1 hybrid individuals which are obtained in the crosses between *Artemia urmiana* (URM), *A. sinica* (SIN), *A. tibetiana* (TIB), *Artemia* sp. from Kazakhstan (KAZ) and diploid parthenogenetic *Artemia* rare males (PD). Error bars are standard deviations. Asterisks ($P \leq 0.05$) indicate significant differences from 50% sex ratio (Chi-square goodness of fit test was employed).

males respectively; $\chi^2 = 111.25$ and $\chi^2 = 16.49$, 1 df, $p < 0.05$), but was non-significantly different from 50% in the F2 PD × SIN (43% of males; $\chi^2 = 0$, 1 df, $p < 0.05$) (Fig. 3). Furthermore, we observed differences in the sex ratio of the F2 offspring among different pairs from the same cross, in particular for F2 PD × KAZ and F2 PD × URM crosses (see Table 2). In the cross F2 PD × KAZ, which higher sample sizes, one pair produced offspring with an even sex ratio (pair 3) while the remaining five pairs had were female biased offspring (see Table 2).

**Table 2 Sex ratio and parthenogenetic females found in F2 PD × KAZ, F2 PD × URM and F2 PD × SIN *Artemia* offspring.** Asterisks ($P \leq 0.05$) indicate significant differences from 50% sex ratio (number of males/total individuals) (Chi-square goodness of fit test was employed). All females obtained were isolated until their deaths to determine their mode of reproduction.

| | Pair | Females | Males | Total | Sex ratio (%) | Parthenogenetic females/analysed females | Parthenogenetic females (%) |
|---|---|---|---|---|---|---|---|
| **F2 PD × KAZ** | *1* | 10 | 0 | 10 | 0.00** | 3/10 | 30 |
| | *2* | 10 | 2 | 12 | 16.67* | 1/10 | 10 |
| | *3* | 7 | 8 | 15 | 53.33 | 0/7 | 0 |
| | *4* | 20 | 0 | 20 | 0.00** | 6/10 | 60 |
| | *5* | 68 | 2 | 70 | 2.86** | 2/4 | 50 |
| | *6* | 31 | 1 | 32 | 3.13** | – | – |
| **Total** | | 146 | 13 | 159 | | 12/41 | 29.27 |
| **F2 PD × URM** | *1* | 16 | 3 | 19 | 15.79** | 0/16 | 0 |
| | *2* | 2 | 4 | 6 | 66.67 | 0/2 | 0 |
| | *3* | 2 | 0 | 2 | 0.00 | 0/2 | 0 |
| | *4* | 3 | 1 | 4 | 25.00 | 1/3 | 33.33 |
| | *5* | 2 | 1 | 3 | 33.33 | – | – |
| | *6* | 2 | 0 | 2 | 0.00 | – | – |
| | *7* | 13 | 2 | 15 | 13.37** | 1/13 | 7.69 |
| **Total** | | 40 | 11 | 51 | | 2/36 | 5.56 |
| **F2 PD × SIN** | *1* | 15 | 13 | 28 | 46.43 | 0/15 | 0 |
| | *2* | 13 | 24 | 37 | 64.86 | 0/13 | 0 |
| | *3* | 6 | 3 | 9 | 33.33 | 0/6 | 0 |
| | *4* | 1 | 3 | 4 | 75.00 | 0/1 | 0 |
| | *5* | 14 | 12 | 26 | 46.15 | 0/14 | 0 |
| | *6* | 10 | 10 | 20 | 50.00 | 0/10 | 0 |
| | *7* | 20 | 18 | 38 | 47.37 | 0/20 | 0 |
| | *8* | 23 | 24 | 47 | 51.06 | 0/23 | 0 |
| | *9* | 30 | 41 | 71 | 57.75 | 0/30 | 0 |
| | *10* | 5 | 8 | 13 | 61.54 | 0/5 | 0 |
| | *11* | 16 | 0 | 16 | 0.00** | 0/16 | 0 |
| | *12* | 7 | 0 | 7 | 0.00** | 0/7 | 0 |
| | *13* | 4 | 1 | 5 | 20.00 | 0/4 | 0 |
| | *14* | 14 | 21 | 35 | 60.00 | 0/14 | 0 |
| **Total** | | 178 | 178 | 356 | | 0 | 0 |

## Generation of hybrid parthenogenetic individuals

Some females isolated from males of all F2 hybrid offspring analysed (when males were present) reproduced parthenogenetically in two of the three crosses. Specifically, 12 out of 41 isolated females (29.27%) were parthenogenetic in F2 PD × KAZ (four out of the five offspring analysed, Table 2), and two out of 36 (5.56%) isolated females in F2 PD × URM (two of five offspring analysed, Table 2). The percentages of parthenogenetic female offspring in the F2 crosses were significantly different from each other ($\chi^2 = 7.24$,

1 df, $p < 0.05$), and only that one in F2 PD $\times$ KAZ did not differ significantly from the expectations of 25% ($\chi^2 = 0.4$, 1 df, $p > 0.05$) under expectations of a recessive allele in a single locus determining parthenogenesis. In all but one case, parthenogenetic females were produced in offspring with significantly female-biased sex ratios (Table 2). None of the 21 F2 PD $\times$ SIN offspring included females that could reproduce parthenogenetically.

### Paternity analysis

In order to examine the parentage of newly generated hybrid parthenogenetic individuals we integrated the information from the mitochondrial COI and from microsatellites markers (Table 3). Six of the 10 analysed females from pair 4 of the cross F2 PD $\times$ KAZ were parthenogenetic and produced F3 clones. As expected, all of them shared their mtDNA haplotype with their sexual grandmother, and amplified one paternal allele in the two informative microsatellite loci, confirming that they were the offspring of the rare male used in the crosses. The F3 generation was overall composed by females and by two rare males with the same genotype as their F2 mothers.

The F2 offspring of two pairs from the crosses PD $\times$ URM (pairs 4 and 7), composed of three and 13 females respectively, included a parthenogenetic female that produced F3 parthenogenetic clones. In both cases, the F2 parthenogenetic female shared its COI haplotype with its sexual grandmother. In one cross, one paternal allele was detected in the F2 hybrid female at each of the three microsatellite loci; in the other cross, the parthenogenetic female inherited one paternal allele at the two informative loci. Most individuals of the F3 generation, composed of females and one rare male in both crosses, have the same genotype as their F2 mothers, with a few exceptions that lacked one of the maternal alleles, suggesting some level of recombination consistent with automixis parthenogenesis.

## DISCUSSION

This study reports for the first time the laboratory generation of parthenogenetic *Artemia* lineages through hybridization via rare males, i.e., through contagious parthenogenesis (*Simon et al., 2003*), shedding light on the possible evolutionary role of parthenogenetically produced males and the genetic basis of parthenogenesis in this genus.

Contagious parthenogenesis may have important evolutionary consequences as it results in the repeated generation of new asexual genotypes, increasing the genetic diversity in parthenogens. This may counteract the loss of asexual genotypes resulting from the accumulation of deleterious mutations (Muller's ratchet) or gene conversion (*Tucker et al., 2013*) and could contribute to the evolutionary success of parthenogenesis (*Simon et al., 2003*).

The occurrence of contagious parthenogenesis relies on regular or occasional hybridization with absence of complete reproductive isolation between parthenogenetically produced males and closely related sexual females (*Simon et al., 2003*). In a previous study, we showed the absence of prezygotic isolation between rare males and Asiatic sexual *Artemia* species since these males often coexist in the same environment of a sexual species (*Abatzopoulos et al., 2006*; *Agh et al., 2007*; *Agh et al., 2009*; *Shadrin,*

Table 3 **Mitochondrial cytochrome c oxidase subunit I (COI) and microsatellite loci analyses for parental individuals (F0) and for parthenogenetic F2 and F3 offspring obtained from the hybrid *Artemia* crosses.** Genotypes for three microsatellite loci (allele sizes in base pairs) are shown. Diagnostic alleles, that is, alleles present in the rare male grandfather and not in the grandmother are shown in bold in the grandfather and in the F2 and F3 offspring. 'Ø' indicates the presence of null alleles; 'm' indicates a rare male. COI haplotypes as named in GenBank are shown. KAZSEX03: GU591387; APD02: DQ426825; AUKOY02: KF707698; AUKOY01: KF707699.

| | Sample code | Apd02 | Apd03 | Apd05 | COI |
|---|---|---|---|---|---|
| *Rare male x*<br>*Artemia sp. Kazakhstan* | F0 (F-Kaz 8) | 233-233 | 213-245 | Ø-Ø | KAZSEX03 |
| | F0 (M-Iraq 8) | 233-**242** | 208-**231** | 115-Ø | APD02 |
| | F2-8-2-2 | 233-233 | **231**-245 | Ø-Ø | KAZSEX03 |
| | F2-8-2-3 | 233-**242** | **231**-245 | Ø-Ø | KAZSEX03 |
| | F2-8-2-4 | 233-**242** | **231**-245 | Ø-Ø | KAZSEX03 |
| | F2-8-2-5 | 233-**242** | **231**-245 | Ø-Ø | KAZSEX03 |
| | F2-8-2-6 | 242-**242** | **231**-245 | Ø-Ø | KAZSEX03 |
| | F2-8-2-8 | 233-**242** | **231**-245 | Ø-Ø | KAZSEX03 |
| | F3-8-2-2-3 | 233-233 | **231**-245 | Ø-Ø | KAZSEX03 |
| | F3-8-2-2-5 | 233-233 | **231**-245 | Ø-Ø | KAZSEX03 |
| | F3-8-2-2-10 | 233-233 | **231**-245 | Ø-Ø | KAZSEX03 |
| | F3-8-2-2-12m | 233-233 | **231**-245 | Ø-Ø | KAZSEX03 |
| | F3-8-2-6-3 | **242**-242 | **231**-245 | Ø-Ø | KAZSEX03 |
| | F3-8-2-6-4 | **242**-242 | **231**-245 | Ø-Ø | KAZSEX03 |
| | F3-8-2-6-5 | **242**-242 | **231**-245 | Ø-Ø | KAZSEX03 |
| | F3-8-2-6-7m | **242**-242 | **231**-245 | Ø-Ø | KAZSEX03 |
| | F3-8-2-8-1 | 233-**242** | **231**-245 | Ø-Ø | KAZSEX03 |
| | F3-8-2-8-2 | 233-**242** | **231**-245 | Ø-Ø | KAZSEX03 |
| | F3-8-2-8-3 | 233-**242** | **231**-245 | Ø-Ø | KAZSEX03 |
| | F3-8-2-8-4 | 233-**242** | **231**-245 | Ø-Ø | KAZSEX03 |
| *Rare male x A. urmiana* | F0 (F-Koy 15) | 233-281 | 207-Ø | 170-Ø | AUKOY02 |
| | F0 (M-Iraq 15) | **254**-233 | **216**-231 | 115-**185** | APD02 |
| | F2-15-8-A | **254**-254 | 207-**216** | **185** | AUKOY02 |
| | F3-15-8-A-1 | **254**-254 | **216** | **185** | AUKOY02 |
| | F3-15-8-A-4 | **254**-254 | 207-**216** | **185** | AUKOY02 |
| | F3-15-8-A-5 | **254**-254 | 207-**216** | **185** | AUKOY02 |
| | F3-15-8-A-6 | **254**-254 | 207-**216** | **185** | AUKOY02 |
| | F3-15-8-A-7m | **254**-254 | 207 | **185** | AUKOY02 |
| *Rare male x A. urmiana* | F0 (F-Koy 16) | 248-Ø | 208-Ø | 90-90 | AUKOY01 |
| | F0 (M-Iraq 16) | 233-**251** | 216-230 | **117**-189 | APD02 |
| | F2-16-7-4 | 248-**251** | Ø-Ø | 90-**117** | AUKOY01 |

Table 3 (*continued*)

| Sample code | Apd02 | Apd03 | Apd05 | COI |
|---|---|---|---|---|
| **F3-16-7-4-1** | 248-**251** | Ø-Ø | 90-**117** | AUKOY01 |
| **F3-16-7-4-2** | 248-**251** | Ø-Ø | 90-90 | AUKOY01 |
| **F3-16-7-4-3** | 248-**251** | Ø-Ø | 90-**117** | AUKOY01 |
| **F3-16-7-4-5** | 248-**251** | Ø-Ø | 90-**117** | AUKOY01 |
| **F3-16-7-4-7m** | 248-**251** | Ø-Ø | 90-**117** | AUKOY01 |

*Anufriieva & Galagovets, 2012*; *Van Stappen et al., 2007*; *Van Stappen, 2008*; *Zheng & Sun, 2013*), show normal pairing behaviour and are fully functional and capable of fertilizing eggs from females of sexual Asiatic *Artemia* species producing viable hybrid offspring (*Maccari et al., 2013*). Under laboratory conditions, each combination of sexual species with rare males produced morphologically normal, viable sexual hybrid F1. Their survival rate to adulthood was over 50% for all the hybrid populations, a high value if compared to survival of F1 of intraspecific crosses of the different *Artemia* species (*Browne & Wanigasekera, 2000*).

We found that females constitute approximately 50% of each F1 hybrid population, an even sex ratio that usually characterizes *Artemia* sexual populations, and this was confirmed by their inability to reproduce without males. These results ruled out a dominant gene as the genetic basis of parthenogenesis. Although all laboratory F1 lines were found to combine ovoviviparous and oviparous reproduction, we observed a strong reduction in the reproductive output in all crosses when compared with the reproductive performance of the parental crosses (*Maccari et al., 2013*). Ovoviviparous broods were mostly made up by abortive embryos (more than 80%) in all the crosses and live nauplii represented only 25% of the offspring in the cross F2 PD × SIN, and less than 10% in all the other crosses (F2 PD × KAZ, F2 PD × URM and F2 PD × TIB). Oviparity, the production of dormant encysted embryos that are resistant to extreme environmental conditions, was represented by a variable quantity of properly shelled embryos, only 25% in the F2 PD × TIB increasing up to 61% in F2 PD × URM. Similarly, a decline in nauplii F2 production occurs in the interspecific crosses between *A. tibetiana* and *A. sinica* (*Van Stappen et al., 2003*).

In contrast to the high survival rates of F1 hybrids, hybrid breakdown was evident in the F2 generation. Nauplii from the F2 generations had low survival rates and were completely inviable in the F2 PD × TIB generation. The lower fertility level of F1 laboratory populations and the reduced viability of F2 hybrid individuals suggest partial genetic incompatibility between parthenogenetic males and sexual females. However, the production of some viable offspring both in F1 and F2 in all hybrid crosses is not so surprising given the recent evolutionary origin of diploid parthenogenetic lineages (Holocene) (*Muñoz et al., 2010*; *Maccari, Amat & Gómez, 2013*).

In two of the three F2 generations (F2 PD × KAZ and F2 PD × URM) we identified 14 hybrid females that upon reaching maturity were capable of parthenogenetic reproduction. Surprisingly, these parthenogenetic females were produced by pairs yielding

strongly female biased F2 offspring. Genetic analysis confirmed the parentage of the parthenogenetic lineages found as the F2 individuals inherited the COI haplotype from the sexual grandmother but included some paternal alleles at nuclear markers, showing that they were the offspring of the rare male used in the crosses. Our results contrast with previous observations suggesting that rare males in the genus *Artemia* are not capable to transmit parthenogenesis-inducing alleles (*Bowen et al., 1978*).

The production of parthenogenetic individuals only in the second generation, suggests that the parthenogenesis-inducing alleles are recessive in *Artemia*. A single-locus recessive inheritance of obligate parthenogenesis also occurs in *Apis mellifera capensis* and in *Lysiphlebus fabarum* (*Sandrock & Vorburger, 2011*; *Lattorff, Moritz & Fuchs, 2005*; *Lattorff et al., 2007*). This is in contrast with *D. pulex*, where the sex-limited meiosis suppression genes are dominant and the asexual clones arise in the first generation (*Innes & Hebert, 1988*). If a single recessive locus was responsible for parthenogenesis and there was no differential viability in *Artemia*, a 25% of parthenogenetic females would be expected in the F2 generation. The proportion of isolated females that reproduced parthenogenetically differed between the crosses. In the cross F2 PD × KAZ, the overall proportion of parthenogenetic F2 females was 29.27%, not significantly different from 25%, whereas in the cross F2 PD × URM this was much lower (5.56%) and significantly different from the expectations for a single recessive locus. These results suggest either differences in the mechanism underlying parthenogenesis between populations, or increased incompatibilities between PD and URM resulting in viability differences linked to the putative locus associated to parthenogenesis. The latter is supported by the lower viability of F2 PD × URM nauplii. The finding of parthenogenetic females only in sex-biased broods suggests that the inheritance of parthenogenesis has a more complex genetic basis, however. Given that females are heterogametic (WZ) (*Bowen, 1963*; *Bowen, 1965*; *Stefani, 1963*) and that F1 females are sexual, we can rule out complete sex-linkage (Z-linkage) of the parthenogenesis determining gene, otherwise parthenogenesis should be apparent in the F1, given that all F1 females are WZ with their Z chromosome presumably inherited from their asexual father. Sex-biased sex ratios are not uncommon in hybrid offspring and can be due to the evolution of sex-ratio distorters and counter evolution of suppressor genes in different lineages (*Hurst & Pomiankowski, 1991*). Our data suggests an interaction between a sex ratio distorter (possibly sex-linked) and a parthenogenetically determining factor. Alternatively, the same gene determining parthenogenesis could act as a sex ratio distorter in heterozygous F1 females, increasing the likelihood of transmission of the W chromosome. Our results do not support differential male mortality, as there was no correlation between brood survival and sex ratio (data not shown). These interpretations must be taken with caution given the limitations of our experimental design and data, as we analysed F2 broods where there was a larger number of nauplii, the survival of the F2 was low, and we cannot rule out some effect of differential sterility. These factors might have biased our conclusions regarding the genetic basis of parthenogenesis. Therefore, to fully understand the genetic basis of parthenogenesis in *Artemia* additional crosses and a large set of marker loci will be necessary.

The ability of sexual females of *A. urmiana* and *Artemia* sp. from Kazakhstan to generate parthenogenetic clones when crossed with rare males is not surprising, as the two main mitochondrial haplogroups of diploid parthenogenetic *Artemia* lineages are related to these species (*Muñoz et al., 2010*; *Maniatsi et al., 2011*; *Maccari, Amat & Gómez, 2013*). However, the more distantly related *A. sinica* (*Baxevanis, Kappas & Abatzopoulos, 2006*; *Hou et al., 2006*) did not produce any parthenogenetic offspring, despite high survival rate in the F2, suggesting that the specific genomic background affect the expression of the gene inducing parthenogenesis. Although repeated gene flow between sexual females and asexual males through contagious parthenogenesis would be expected to result in a regular emergence of asexual strains with diverse maternal origins, the fact that just two, possibly three, maternal origins of parthenogenetic lineages have been identified (*Muñoz et al., 2010*; *Maniatsi et al., 2011*; *Maccari, Amat & Gómez, 2013*) indicate that the incidence of contagious parthenogenesis, if this is the mechanism of origin, must be extremely low in natural environments. Indeed, the rare males must be present in the population at the same time as the sexual females of the related species, and given that both parthenogenetic and sexual species often have different ecological requirements, they may overlap just during part of each season (*Amat et al., 1991*; *Ghomari et al., 2011*). In addition, the percentage of rare male production by diploid parthenogenetic females is very low, about 1–16 in 1000 (*Maccari et al., 2013*). Then, as the parthenogenesis occurs in the second generation (i.e., is based on a recessive trait), a F1 × F1 mating must occur for parthenogenesis to appear in the offspring. Finally, F2 survival is very reduced, overall making the origin of a parthenogenetic lineage an unlikely event in the wild.

Our study is the first to generate new parthenogenetic lineages in *Artemia* by mating rare males from parthenogenetic genotypes with sexual females, providing evidence that contagious parthenogenesis can potentially occur in the genus *Artemia*. This conclusion does not rule out that other mechanisms (spontaneous origin or hybridisation) might have been also responsible for the origin of parthenogenetic lineages. Demonstration of contagious parthenogenesis as the mechanism underlying parthenogenesis in *Artemia* in the wild will necessitate the use of genomic tools. Further studies on hybrid fitness would be necessary to estimate the strength of reproductive isolation and to compare the reproductive performance of laboratory-produced parthenogenetic clones with the parental parthenogenetic strains. The origin of independently reproducing parthenogenetic clones in the laboratory raises the question of the survival of these clones when competing with sympatric sexual species.

Given that many parthenogenetic organisms produce males occasionally (*van der Kooi & Schwander, 2014*) and such males are still able to maintain their functionality, the occurrence of contagious parthenogenesis could be more widespread than currently acknowledged.

## ACKNOWLEDGEMENTS

We wish to thank Paul Nichols for his help with microsatellite screening and Mónica Barbosa, Eva Becerro and Diana Guinot for their help with the laboratory experiments.

We thank Maria José Carmona for her constructive suggestions on a previous version of this manuscript. We thank the editor Tanja Schwander, and David Innes and two anonymous reviewers for their constructive comments that substantially improved the manuscript.

### Funding

This study was funded by the Plan Nacional CGL2008-03277 project to FA, sponsored by the Spanish Government MICIN. AG was supported by a National Environment Research Council (NERC) Advanced Fellowship (NE/B501298/1). MM was supported by a fellowship of the JAE Program from CSIC and European Social Fund. The funders had no role in study design, data collection and analysis, decision to publish, or preparation of the manuscript.

### Grant Disclosures

The following grant information was disclosed by the authors:
MICIN Plan Nacional: CGL2008-03277.
National Environment Research Council (NERC) Advanced Fellowship: NE/B501298/1.
JAE-CSIC.
European Social Fund.

### Competing Interests

The authors declare there are no competing interests.

### Author Contributions

- Marta Maccari conceived and designed the experiments, performed the experiments, analyzed the data, wrote the paper, prepared figures and/or tables, reviewed drafts of the paper.
- Francisco Amat conceived and designed the experiments, contributed reagents/materials/analysis tools, reviewed drafts of the paper.
- Francisco Hontoria contributed reagents/materials/analysis tools, reviewed drafts of the paper.
- Africa Gómez conceived and designed the experiments, contributed reagents/materials/analysis tools, wrote the paper, reviewed drafts of the paper.

### Supplemental Information

Supplemental information for this article can be found online at http://dx.doi.org/10.7717/peerj.439.

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
