# Peer review of "Laboratory generation of new parthenogenetic lineages supports contagious parthenogenesis in Artemia"

_PeerJ, doi:10.7717/peerj.439_

## Round 0.1 · original submission · Minor Revisions

In your revision, please pay particular attention to the following points:

1) Provide a more quantitative analysis of the genetic architecture of parthenogenesis in Artemia (number of loci, dominance etc), including a discussion of female heterogamety in this system. This would also allow you to more clearly highlight the 'novel' findings of the present ms relative to the reports of functional males produced by parthenogenetic females in your 2013 publication in JEB.

2) I think you could extend your discussion on whether the different parthenogenetic Artemia lineages were likely generated via contagious parthenogenesis, or whether they evolved independently from each other. This would of course depend on the relative fitnesses of sexual, parthenogenetic, and different sex-parthenogenetic hybrid females, but also on how likely 'parthenogenetically-produced males' are to encounter sexual females in natural populations and father offspring when in competition with sexual males. Overall, you seem to assume that because 'parthenogenetically-produced' males are functional, their production is adaptive. However, whether this is the case depends on the fitness of these males in natural populations and whether or not the production of such males is costly for females.

Reviewer 1 ·

Basic reporting

No Comments

Experimental design

No Comments

Validity of the findings

No Comments

Additional comments

Original review :

The authors investigated the occurrence of contagious asexuality from the asexual species A. parthenogenetica to its Asian sexual relatives. Parthenogenetically produced rare males were mated with females from three closely related sexual species. No F1 female was found to reproduce parthenogenetically when isolated. However, some of the F2 females (offspring from a mating between F1 siblings) were found to be parthenogenetic. Hence this study is one of the few studies demonstrating contagious asexuality and is the first to extensively investigate the phenomenon in the Artemia genus. I found the study very interesting as it sheds new lights on the genetic determinism of parthogenesis in general and in the Artemia genus in particular.
I was a bit disapointed by the qualitative rather than quantitative tone of the paper. The genetic determinism of parthenogenesis is usually difficult to investigate as genetic crosses are impossible to conduct in most asexual species. The demonstration that rare males can transmit genes for parthenogenesis is thus very interesting ! I hope the major and minor comments below will be helpful in revising the manuscript.

Major comments:

I thought the genetic interpretation of the results could be more thorough. This would strengthen the paper and make it less anecdotal. Indeed, I felt that the authors emphasized the novelty of the synthesis of parthenogenetic lineages in the lab in the abstract, introduction and discussion; at the expense of discussing the general implications of their work. For example, they missed the interesting point of formally testing hypotheses regarding the number of loci involved in parthenogenesis. A suggestion would be: 1/ To mention or comment on the cytological observations by R. Stefani (Stefani, R., 1963: La digametia femminile in Artemia salina Leach e la constituzione del corredo cromosomico nei biotipi diploide anfigonico e diploide partenogenetico. Caryologia, 16: 629-636) that diploid A. parthenogenetica females are heterogametic (ZW). I think the authors implicitly assume that the rare males are ZZ when testing for a biased sex ratio (L 108, L228).
2/ Explicitly mention in the introduction the different hypotheses (ie dominant or recessive locus/loci for parthenogenesis). The protocol used to test these hypotheses (eg. dominant locus =50% of parthenogenetic F1 females, recessive locus= 25% of parthenogenetic F3 females) could be presented in the methods. Note that the test for a dominant locus should not be presented in the results, L159-160. Similarly the expectation for a recessive locus should be mentioned before the discussion (L259).
2/ Add a column in Table 2 with the proportion of parthenogenetic F3 females. Different hypotheses could then be formally tested (eg. One locus vs. two unlinked loci determining parthenogenesis, etc…).

Of course genetic incompatibilities between A. parthenogenetica and the different sexual species might complicate this scenario. The authors could mention that A. sinica is the most distantly related sexual species (Hou, L., Bi, X., Zou, X., He, C., Yang, L., Qu, R., Liu, Z., 2006. Molecular systematics of bisexual Artemia populations. Aquaculture Research 37, 671-680. ; Baxevanis, A.D., Kappas, I., Abatzopoulos, T.J., 2006. Molecular phylogenetics and asexuality in the brine shrimp Artemia. Molecular phylogenetics and evolution 40, 724 - 738.). Hence, A. sinica represents some kind of ‘outgroup’ here. So it would be interesting to interpret the surprinsing result that F2 offspring from crosses with A. sinica have the highest survival (L 154-155). The authors should mention the survival of their controls (ie is this observation only due to maternal effects with A. sinica females producing better quality offspring on average ?). The difference in biased sex ratio between F2 offspring from crosses from A. urmiana, A. sp Kazakhstan and A. sinica could also be discussed in more details. Is-there some kind of species-specific meiotic drive where F1 females preferentially transmit the Z from the rare male compared to the W from the sexual species ? Finally, it is worth noting that none of the F2 females from the cross with A. sinica were parthenogenetic. Similarly, the ‘parthenogenetic’ locus/loci could have a species or ‘background’-specific effect.

The autors should be aware that the view that automictic parthenogenesis is the prominent reproductive mode in Artemia parthenogenetica (L 46) has been questioned by R. Stefani. It should indeed produce homogametic males (ZZ) and females (WW) in equal proportions (Stefani, R., 1960. L'Artemia salina pathenogenetica a Cagliari. Riv. Biol. 53, 463-490, Stefani, R., 1967. La maturazione dell'uovo nell'Artemia salina di Sète. Rivista di Biologia 60, 599-615). In this study, most of the F3 offspring are exactly identical to their parthenogenetic mother for the 3 loci investigated ; which is at odds with the hypothesis of a prominent automixis. The authors could comment on the two exceptions (Apd03 locus in 2 out 5 F3 offspring from the cross with A. urmiana) and discuss the different hypotheses L 204-205 (automixis vs. allele drop out vs. aneuploidy).

I thought that the results from earlier studies could be better presented (ie. Crossed with the distantly related A. franciscana in Bowen et al 1978, MacDonald & Browne 1987, with A. sinica in Cai 1993 and with A. urmiana in Bowen et al 1978). Importantly, the protocol used by S. Bowen on the F1 offspring from the A. urmiana cross could not test the presence of a recessive/dominant locus determining parthenogenesis. Hence, your results do not “disagree” with these previous observations (L252-253). These previous observations just didn’t use the right protocol.

Minor comments:

L 24-27: This sentence might be unclear to some readers.

L 31: “on chromosome VII”

L 33: “hybridization with”

L 39: “diverse ploidy levels”

L 42: maybe you want to say something like: “where they co-occur with several sexual species in sympatry”

L 61-77: This information could be summarized in a table. The originality of this study could be emphasized by adding a column with the information on the genetic mechanism responsible for parthenogenesis in each example.

L 94: Based on geographical distances, the A. urmiana population from Koyashskoe (Ukraine) might be more distantly related to the Bagdad A. parthenogenetica clone (Irak) than the Urmia A. urmiana population (Iran). Please could you explain why using a population where genetic incompatibilities with A. parthenogenetica might be more important?

L116: how did you determine if a brood was not fertilized? Please mention the criteria used to stop monitoring the focal female.

L118: Did you observe non-dormant embryos that would be laid by the female and hatch outside of the brood pouch? How would these be classified?

L123-125: Were F2 females isolated/followed until they died? It could be worth mentioning that non-reproducing females could either be sexual or sterile (eg. due to some genetic incompatibilities).

L129: This test also allows testing for a loss of heterozygozity between the F2 female and its parthenogenetic F3 offspring. This would give a more positive tone to this section (compared to the more negative checking of a potential contamination).

L140/190-197: This could be also presented as a test of maternal mitochondrial inheritance in the cross (rather than paternal or biparental inheritance).

L142: Please use F0, F1, F2 for consistency.

L175-176: Are these differences significantly different? If they are could you indicate this on Figure 1?

L177-179: Please report the test, P-values and df here.

L181-182: Please provide the observed sex ratios here and the reference for the table.

L185/L246: “hybrid lineage/generations” sounds weird/inappropriate.

L186: Please check the number of broods and refer to the actual table, eg. “(four out of the seven broods analysed, Table 2)”

L192-196: why giving details for one cross?

L198: “each of the seven”?

L204: “same genotype as their F2 mother”

L221: could you actually provide a reference for their sympatric occurrence?

L225: Is adulthood defined as maturity? Please clarify the duration of the monitoring period for survival/parthenogenetic reproduction throughout the manuscript (eg. legend of Table 2: “were isolated until…”, etc..).

L226: ‘good survivorship’ seems quite vague, wouldn’t be a comparison with the survivorship of intraspecific crosses a better comparison (as these would control for differences in lab / maternal effects).

L228: This is again consistent with rare males being homogametic with ZZ chromosomes.

L233-238: Could you interpret this and compared your result with those from other studies performing interspecifics crosses (Van Stappen et al 2003)?

L247-249: Why do you expect a link between a biased sex ratio in the F2 and F2 parthenogenetic reproduction? Do you imply a link between sex chromosomes and the locus/loci determining parthenogenesis?

L261-262: Please mention that these values represent the overall proportion of parthenogenetic F2 females (as opposed to the mean proportion of parthenogenetic F2 females).

L262-264: Please clarify these two alternative hypotheses.

L270-274: As shown by your two previous articles, mitochondrial polymorphism is quite low in A. urmiana and A. sp. Kazakhstan. Hence, the hypothesis of independent origins of multiple clonal lineages with similar mitochondrial haplotypes cannot be ruled out.

L276: “the low percentage of rare male production by diploid parthenogenetic females”

L277: “compared to sexual ones”

L285: “clones in the laboratory raises the question of the survival of these clones when competing with sympatric sexual species”

Figure 2 (legend): “abortive embryos”

Figure 3: Did you perform a test for each individual brood in Table 2 and a test for the overall sex ratio in Fig. 3? Please clarify.

Table1: Did you do a test to see if the proportion of fertilized broods differed between the four types of crosses?

Table2: Please use “.” as a decimal separator.

Table 3: The relationship between the F2 female and its offspring could be clearer. Maybe you could add another column with the generation (F0, F1 and F2) and have the F3 offspring below their respective mothers.

Maybe you could merge Supp. Table S2 and S3 together. Be careful with the translation of the column headers in Supp. Table 3.

Reviewer 2 ·

Basic reporting

In this paper, the authors scrutinise the question of whether rare sons produced by thelytokous populations of the shrimp Artemia have the potential to start new parthenogenetic lineages by mating with sexual females. This is an interesting question and the paper is generally well written.

However, I have one major concern (and a few minor ones listed below) that needs to be addressed before publication. In their previous study (Maccari et al. 2013, JEB), the authors had already shown that males produced by parthenogenetic females can mate with and produce viable offspring with asexual females. In the first section of the present paper, the same kind of results are reported again, but it's not clear whether a new experiment was conducted here (and if so, whether this was just a replicate or somehow different), or whether merely the old results were recapitulated. This apparent overlap between the two articles needs to be clarified.
In addition, given that F1's had been generated and thus functionality of asexually produced males already confirmed previously, I find the scope of this study quite narrow (essentially assessing reproductive performance of F1 and producing and characterising F2). I would therefore suggest to go one step further and assess different fitness components of the F3 parthenogenetic females and compare them to those of their parthenogenetic great-grandmothers. Even better would be to continue the experiment for a few more generations to see if new asexual lineages can really be established. This would then help answer the important question of why, if contagious parthenogenesis is possible, only very few parthenogenetic lineages have arisen in this species complex.


Minor comments & typos:

-line 245: "holocene" is a bit vague. Is there no concrete estimate for the age of those lineages?

- line 260-262 (and lines 184-188): I would like to see some statistical analysis concerning this important point. Are these two proportions significantly different from each other, and are they significantly different from the expectation of 25% under the hypothesis of a recessive allele inducing parthenogenesis?

- Figure 2: I find this figure confusing because the percentages for each cross do not add up to 100%. Please clarify.

- Table 2: Decimal separators needs to be full stops rather than commas
line 41: "where THEY co-occur"
line 210: "shedding light ON the"
line 277: "COMPARED to sexual ones"
line 378: Reference incomplete

Experimental design

No comments.

Validity of the findings

No comments.

·

Basic reporting

The manuscript is clearly written and conforms to all of the PeerJ policies. The figures and tables are relevant to the content of the manuscript.

Experimental design

Some parthenogenetic Artemia can produced rare males that can potentially mate with sexual females and generate new parthenogenetic genotypes. The manuscript reports results from such laboratory crosses confirming the origin of new parthenogenetic genotypes by this mechanism. Therefore, these results support a model for the contagious spread of parthenogenetic lineages in Artemia.

The present study took advantage of crosses from a previous study (Maccari et al. 2013) that produced F1 hybrids between the rare males and sexual females. In the present study F2 progeny were produced by mating 24 male and females pairs from within each F1 family. Of the 77 female F2 offspring that were reared in isolation, 14 showed the ability to reproduce parthenogenetically.

Validity of the findings

Microsatellite and mtDNA genetic markers were used to confirm maternity and paternity, supporting the transmission of genes by the rare males produced by parthenogenetic females. The data confirm the generation of new parthenogenetic genotypes from crosses involving rare males with sexual Artemia females.

Additional comments

Throughout the manuscript it would be better to refer to mating between rare males from parthenogenetic Artemia with sexual females (resulting in new parthenogenetic lineages) as supporting the contagious parthenogenesis model for the spread of asexuality. Suggested re-wordings follow:

Title: "Laboratory generation of new Artemia parthenogenetic lineages through contagious parthenogenesis"

Change to: "Laboratory generation of new parthenogenetic lineages supports contagious parthenogenesis in Artemia"

Abstract: "Here we investigate whether new parthenogenetic clones could be obtained in the laboratory through contagious origin."

Change to: "Here we investigate whether new parthenogenetic clones could be obtained in the laboratory by mating these rare males with sexual females."

Abstract: "Our study documents the first synthesis of parthenogenetic lineages through contagious parthenogenesis in Artemia."

Change to: "Our study documents the first laboratory synthesis of new parthenogenetic lineages in Artemia and supports a model for the contagious spread of parthenogenesis."

Abstract: "We discuss the possible genetic mechanisms responsible for parthenogenesis and the likelihood of contagious parthenogenesis in natural environments."

Avoid "We discuss" and state that the results confirm recessive inheritance and further experiments are required to confirm the likelihood of the contagious parthenogenesis model.

Line 41 "Parthenogenetic populations are restricted to the Old World where co-occur with several sexual species"

insert "they" after "where"

Line 78 "Here we assess the reproductive role of Artemia rare males investigating whether new parthenogenetic clones could arise in laboratory through contagious origin."

Change to: "Here we assess the reproductive role of rare males and investigate whether new parthenogenetic clones could be produced in the laboratory as support for the contagious origin of parthenogenetic lineages in Artemia."

Line 187 "None of the 21 F2 PD x SIN broods included females that could reproduce in isolation."

Change to: "None of the 21 F2 PD x SIN broods included females that could reproduce parthenogenetically."

Line 192 "Six of the 10 analysed females from brood 4 of the cross F2 PD x KAZ were parthenogenetic and produced F3 clones."

If the parthenogenetic females reproduce by automixis, and the F2 are 231-245 heterozygotes for the Apd03 locus, wouldn't some homozygotes be expected in the F3 progeny rather than observing all 231-245 heterozygotes?

Line 196 "... the same genotype than their F2 mothers."

Change to: "...the same genotype as their F2 mothers."

Line 198 "Each of the two analysed F2 broods from the crosses PD x URM (broods 4 and 7), composed by three and 13 females respectively,..."

Change "...composed by three..." to "...composed of three..."

Line 203 "Most individuals of the F3 generation, composed by females and one rare male in both crosses.."

Change "composed by" to "composed of"

Line 220 "...since these males often coexist in the same environment of a sexual species..."

Do the parthenogenetic lineages that produce rare males coexist in the same area and the same habitat as the sexual females? How likely is it that such matings can occur in nature?


Line 241 Change "unviable" to "inviable"

Line 246 "In two of three hybrid generations, F2 PD x KAZ and F2 PD x URM, we identified 14
hybrid females..."

Change to: "In two of the three F2 hybrid generations (F2 PD x KAZ and F2 PD x URM) we identified 14 hybrid females..."

Line 274: "...the incidence of contagious parthenogenesis must be extremely low in natural environments. This could be due to the low percentage of male offspring of parthenogenetic females (Maccari et al., 2013a) or to the possibly lower fitness of newly emerging asexual strains comparing to sexual ones."

What role might recessive inheritance of the allele for parthenogenesis play for contagious spread?

Line 278 "Our findings document the first generation of parthenogenetic lineages through contagious parthenogenesis in Artemia, providing evidence..."

Change to: "Our study is the first to generate new parthenogenetic lineages in Artemia by mating rare males from parthenogenetic genotypes with sexual females, providing evidence..."

Line 283: "...to compare the reproductive performance of contagious parthenogenetic clones with..."

Change to: "...to compare the reproductive performance of laboratory-produced parthenogenetic clones with..."

Line 285 Change "...raises the question either these clones..." to "...raises the question of whether these clones..."

---

## Round 0.2 · accepted · Accept

Dear Mrs Maccari,
Thank you for submitting a revised version of your manuscript, and for carefully considering the comments of the reviewers. Given your revisions, we are happy to accept your ms in its current version.